# Judging LLM-as-a-Judge
# with MT-Bench and Chatbot Arena

Lianmin Zheng[1*]    Wei-Lin Chiang[1*]    Ying Sheng[4*]    Siyuan Zhuang[1]

Zhanghao Wu[1]    Yonghao Zhuang[3]    Zi Lin[2]    Zhuohan Li[1]    Dacheng Li[13]

Eric P. Xing[35]    Hao Zhang[12]    Joseph E. Gonzalez[1]    Ion Stoica[1]

[1] UC Berkeley    [2] UC San Diego    [3] Carnegie Mellon University    [4] Stanford    [5] MBZUAI

## Abstract

Evaluating large language model (LLM) based chat assistants is challenging due to their broad capabilities and the inadequacy of existing benchmarks in measuring human preferences. To address this, we explore using strong LLMs as judges to evaluate these models on more open-ended questions. We examine the usage and limitations of LLM-as-a-judge, including position, verbosity, and self-enhancement biases, as well as limited reasoning ability, and propose solutions to mitigate some of them. We then verify the agreement between LLM judges and human preferences by introducing two benchmarks: MT-bench, a multi-turn question set; and Chatbot Arena, a crowdsourced battle platform. Our results reveal that strong LLM judges like GPT-4 can match both controlled and crowdsourced human preferences well, achieving over 80% agreement, the same level of agreement between humans. Hence, LLM-as-a-judge is a scalable and explainable way to approximate human preferences, which are otherwise very expensive to obtain. Additionally, we show our benchmark and traditional benchmarks complement each other by evaluating several variants of LLaMA and Vicuna. The MT-bench questions, 3K expert votes, and 30K conversations with human preferences are publicly available at `https://github.com/lm-sys/FastChat/tree/main/fastchat/llm_judge`.

## 1 Introduction

There has been a proliferation of LLM-based chat assistants (chatbots) that leverage supervised instruction fine-tuning and reinforcement learning with human feedback (RLHF) to unlock new instruction following and conversational abilities [31, 2, 30, 8, 52, 48, 14]. Once aligned with humans, these chat models are strongly preferred by human users over the original, unaligned models on which they are built. However, the heightened user preference does not always correspond to improved scores on traditional LLM benchmarks – benchmarks like MMLU [19] and HELM [24] cannot effectively tell the difference between these aligned models and the base models. This phenomenon suggests that there is a fundamental discrepancy between user perceptions of the usefulness of chatbots and the criteria adopted by conventional benchmarks.

We argue that this discrepancy primarily arises due to existing evaluation that only measures LLMs' core capability on a confined set of tasks (e.g., multi-choice knowledge or retrieval questions), without adequately assessing its alignment with human preference in open-ended tasks, such as the ability to accurately adhere to instructions in multi-turn dialogues. As a demonstration, we show

---

*Joint first authors. This paper is an extended version of our earlier blog post [8].

37th Conference on Neural Information Processing Systems (NeurIPS 2023) Track on Datasets and Benchmarks.

---
**Question:** If the FED buys bonds in the secondary market
(A) the money supply will increase. (B) the money supply will decrease. (C) the money supply will not be affected. (D) the discount rate would be affected. Output the answer only.

**Assistant A:**

```
The Federal Reserve buys
bonds in the secondary
market to increase the
money supply.
```

**Assistant B:**

```
(A) the money supply will increase.
```

---
**Follow-up Question:** How does it affect my daily life? Give 3 examples.

**Assistant A:**

```
The Federal Reserve buys
bonds in the secondary
market to increase the
money supply. This
affects your daily life
by:
1. Increasing the money
supply.
2. Increasing the money
supply and lowering the
discount rate.
3. Increasing the money
supply and lowering the
discount rate and
increasing the money
supply.
```

**Assistant B:**

```
When the Federal Reserve buys bonds in the secondary market, it can have
several effects on the economy that can impact your daily life. Here are
three examples:

1. Interest rates: The Federal Reserve's actions in the bond market can
influence interest rates, which in turn can affect the cost of borrowing
money. For example, if the Federal Reserve buys bonds and drives up demand
for them, interest rates on things like mortgages, car loans, and credit
cards may go up, making it more expensive to borrow money.
2. Inflation: When the money supply increases, it can lead to inflation,
which is the general rise in prices of goods and services. This can impact
your daily life by making it more expensive to buy things like groceries,
gas, and other necessities.
3. Employment: The Federal Reserve's actions in the bond market can also
have an impact on employment. For example, if the Federal Reserve's
actions lead to economic growth and increased demand for goods and
services, it can create more job opportunities and boost employment.
```

---
**GPT-4 Judgment:**

```
Assistant A provided an incorrect response to the user's question about how the Federal Reserve
buying bonds in the secondary market affects daily life. The answer given is repetitive and lacks
clear examples of how the action impacts daily life.
On the other hand, Assistant B provided a relevant and accurate response to the user's question about
the Federal Reserve buying bonds. The answer includes three clear examples of how the action impacts
daily life, such as interest rates, inflation, and employment.

Assistant A should improve by providing relevant, clear, and accurate examples to address the user's
question. Assistant B's response is more helpful, relevant, and detailed.
```

---

Figure 1: Multi-turn dialogues between a user and two AI assistants—LLaMA-13B (Assistant A) and Vicuna-13B (Assistant B)—initiated by a question from the MMLU benchmark and a follow-up instruction. GPT-4 is then presented with the context to determine which assistant answers better.

conversation histories with two models on an MMLU question in Figure 1. The two models are LLaMA-13B [39], a pre-trained base model without fine-tuning, and Vicuna-13B, our fine-tuned model from LLaMA-13B on high-quality conversations (the training details are in Appendix E). Despite the base LLaMA models showing competitive performance on conventional benchmarks (Table 8), its answers to open-ended questions are often not preferred by humans. This misalignment of conventional benchmarks underscores the core problem driving this paper: *the need for a robust and scalable automated method to evaluate LLM alignment with human preferences.*

To study this, we introduce two benchmarks with human ratings as the primary evaluation metric: MT-bench and Chatbot Arena. MT-bench is a series of open-ended questions that evaluate a chatbot's multi-turn conversational and instruction-following ability – two critical elements for human preference. MT-bench is also carefully constructed to differentiate chatbots based on their core capabilities, such as reasoning and math. In addition, we develop Chatbot Arena, a crowdsourced platform featuring anonymous battles between chatbots in real-world scenarios – Users engage in conversations with two chatbots at the same time and rate their responses based on personal preferences.

While human evaluation is the gold standard for assessing human preferences, it is exceptionally slow and costly. To automate the evaluation, we explore the use of state-of-the-art LLMs, such as GPT-4, as a surrogate for humans. Because these models are often trained with RLHF, they already exhibit strong human alignment. We call this approach *"LLM-as-a-judge"*. This approach has been tried in our earlier blog post [8] and other concurrent or follow-up work [5, 29, 14, 12, 52, 18, 33, 40, 7, 43]. However, there has not been a systematic study of this approach.

In this paper, we study the LLM-as-a-judge approach by comparing it to the gold standard of human evaluation. We examine several potential limitations of the LLM-as-a-judge approach including position bias, verbosity bias, self-enhancement bias, and limited reasoning ability. We show that some of the biases are minor or can be mitigated. Once addressed, our results from 3K controlled expert votes and 3K crowdsourced human votes in the wild verify that GPT-4 judge match

human evaluations at an agreement rate exceeding 80%, achieving the same level of human-human agreement (§4.2, Table 4). Consequently, this suggests LLM-as-a-judge is a scalable method to swiftly evaluate human preference, serving as a promising alternative to traditional human evaluations.

This paper makes two contributions: (1) a systematic study of LLM-as-a-judge; and (2) human preference datasets with high-quality questions and diverse user interactions from MT-bench and Chatbot Arena. In addition, we argue for the adoption of a hybrid evaluation framework for future LLM benchmarks: by combining the existing capability-based benchmarks and the new preference-based benchmarks with LLM-as-a-judge, one can swiftly and automatically evaluate both the core capabilities and human alignment of models. We publicly release 80 MT-bench questions, 3K expert votes, and 30K conversations with human preferences for future study.

Table 1: Sample multi-turn questions in MT-bench.

| Category | | Sample Questions |
|---|---|---|
| Writing | 1st Turn | Compose an engaging travel blog post about a recent trip to Hawaii, highlighting cultural experiences and must-see attractions. |
| | 2nd Turn | Rewrite your previous response. Start every sentence with the letter A. |
| Math | 1st Turn | Given that $f(x) = 4x^3 - 9x - 14$, find the value of $f(2)$. |
| | 2nd Turn | Find $x$ such that $f(x) = 0$. |
| Knowledge | 1st Turn | Provide insights into the correlation between economic indicators such as GDP, inflation, and unemployment rates. Explain how fiscal and monetary policies ... |
| | 2nd Turn | Now, explain them again like I'm five. |

## 2  MT-Bench and Chatbot Arena

### 2.1  Motivation

With the recent advances of LLMs, LLM-based assistants start to exhibit artificial general intelligence across diverse tasks, from writing and chatting to coding [5, 30, 1, 37]. However, evaluating their broad capabilities also becomes more challenging. Despite the availability of numerous benchmarks for language models, they primarily focus on evaluating models on closed-ended questions with short responses. Given that these chat assistants can now precisely follow user instructions in multi-turn dialogues and answer open-ended questions in a zero-shot manner, current benchmarks are inadequate for assessing such capabilities. Existing benchmarks mostly fall into the following three categories.

- **Core-knowledge benchmarks**, including MMLU [19], HellaSwag [50], ARC [9], Wino-Grande [36], HumanEval [6], GSM-8K [10], and AGIEval [51], evaluate the core capabilities of pre-trained LLMs using zero-shot and few-shot benchmark sets. They typically require LLMs to generate a short, specific answer to benchmark questions that can be automatically validated.
- **Instruction-following benchmarks**, such as Flan [27, 46], Self-instruct [44], NaturalInstructions [28], Super-NaturalInstructions [45], expand to slightly more open-ended questions and more diverse tasks and are used to evaluate LLMs after instruction fine-tuning.
- **Conversational benchmarks**, like CoQA [35], MMDialog [15] and OpenAssistant [23], are closest to our intended use cases. However, the diversity and complexity of their questions often fall short in challenging the capabilities of the latest chatbots.

While largely overlooked by existing LLM benchmarks, human preferences serve as a direct measure of a chatbot's utility in open-ended, multi-turn human-AI interactions. To bridge this gap, we introduce two novel benchmarks expressly tailored to assess human preferences. Simultaneously, these benchmarks are designed to distinguish the core capabilities of state-of-the-art models.

### 2.2  MT-Bench

We create MT-bench, a benchmark consisting of 80 high-quality multi-turn questions. MT-bench is designed to test multi-turn conversation and instruction-following ability, covering common use cases and focusing on challenging questions to differentiate models. We identify 8 common categories of user prompts to guide its construction: writing, roleplay, extraction, reasoning, math, coding,

knowledge I (STEM), and knowledge II (humanities/social science). For each category, we then manually designed 10 multi-turn questions. Table 1 lists several sample questions.

## 2.3 Chatbot Arena

Our second approach is Chatbot Arena, a crowdsourcing benchmark platform featuring anonymous battles. On this platform, users can interact with two anonymous models simultaneously, posing the same question to both. They vote for which model provides the preferred response, with the identities of the models disclosed post-voting. After running Chatbot Arena for one month, we have collected around 30K votes. Since the platform does not use pre-defined questions, it allows gathering a wide range of unrestricted use cases and votes in the wild, based on the diverse interests of users. A screenshot of the platform can be found at Appendix C.2.

## 3  LLM as a Judge

While our initial evaluations using MT-bench and Chatbot Arena rely on human ratings, collecting human preferences can be costly and laborious [44, 38, 31, 2, 13]. To overcome this, we aim to develop a more scalable and automated approach. Given that most questions in MT-bench and Chatbot Arena are open-ended without reference answers, devising a rule-based program to assess the outputs is extremely challenging. Traditional evaluation metrics based on the similarity between outputs and reference answers (e.g., ROUGE [25], BLEU [32]) are also ineffective for these questions.

As LLMs continue to improve, they show potential in replacing human annotators in many tasks [17, 20]. Specifically, we are interested in whether LLMs can effectively evaluate the responses of chat assistants and match human preferences. Next, we discuss the use and limitations of LLM-as-a-judge.

### 3.1  Types of LLM-as-a-Judge

We propose 3 LLM-as-a-judge variations. They can be implemented independently or in combination:

- **Pairwise comparison**. An LLM judge is presented with a question and two answers, and tasked to determine which one is better or declare a tie. The prompt used is given in Figure 5 (Appendix).
- **Single answer grading**. Alternatively, an LLM judge is asked to directly assign a score to a single answer. The prompt used for this scenario is in Figure 6 (Appendix).
- **Reference-guided grading**. In certain cases, it may be beneficial to provide a reference solution if applicable. An example prompt we use for grading math problems is in Figure 8 (Appendix).

These methods have different pros and cons. For example, the pairwise comparison may lack scalability when the number of players increases, given that the number of possible pairs grows quadratically; single answer grading may be unable to discern subtle differences between specific pairs, and its results may become unstable, as absolute scores are likely to fluctuate more than relative pairwise results if the judge model changes.

### 3.2  Advantages of LLM-as-a-Judge

LLM-as-a-judge offers two key benefits: *scalability* and *explainability*. It reduces the need for human involvement, enabling scalable benchmarks and fast iterations. Additionally, LLM judges provide not only scores but also explanations, making their outputs interpretable, as shown in Figure 1.

### 3.3  Limitations of LLM-as-a-Judge

We identify certain biases and limitations of LLM judges. However, we will also present solutions later and show the agreement between LLM judges and humans is high despite these limitations.

**Position bias** is when an LLM exhibits a propensity to favor certain positions over others. This bias is not unique to our context and has been seen in human decision-making [3, 34] and other ML domains [22, 41].

Figure 11 (Appendix) shows an example of position bias. GPT-4 is tasked to evaluate two responses from GPT-3.5 and Vicuna-13B to an open-ended question. When GPT-3.5's answer is positioned

Table 2: Position bias of different LLM judges. Consistency is the percentage of cases where a judge gives consistent results when swapping the order of two assistants. "Biased toward first" is the percentage of cases when a judge favors the first answer. "Error" indicates wrong output formats. The two largest numbers in each column are in bold.

| Judge | Prompt | Consistency | Biased toward first | Biased toward second | Error |
|-------|--------|-------------|---------------------|----------------------|-------|
| Claude-v1 | default | 23.8% | **75.0%** | 0.0% | 1.2% |
| | rename | 56.2% | 11.2% | **28.7%** | **3.8%** |
| GPT-3.5 | default | 46.2% | **50.0%** | 1.2% | 2.5% |
| | rename | 51.2% | 38.8% | 6.2% | **3.8%** |
| GPT-4 | default | **65.0%** | 30.0% | 5.0% | 0.0% |
| | rename | **66.2%** | 28.7% | 5.0% | 0.0% |

Table 3: Failure rate under "repetitive list" attack for different LLM judges on 23 answers.

| Judge | Claude-v1 | GPT-3.5 | GPT-4 |
|-------|-----------|---------|-------|
| Failure rate | 91.3% | 91.3% | 8.7% |

Table 4: Judge failure rate on 10 math questions with different prompts. We test LLaMA-13B vs. Vicuna-13B and swap positions. A failure means when GPT-4 says an incorrect answer is correct.

| | Default | CoT | Reference |
|--|---------|-----|-----------|
| Failure rate | 14/20 | 6/20 | 3/20 |

first, GPT-4 considers GPT-3.5's answer more detailed and superior. However, upon switching the positions of the two responses, GPT-4's judgement flips, favoring Vicuna's answer.

To analyze the position bias, we construct two similar answers to each first-turn question in MT-bench by calling GPT-3.5 twice with a temperature of 0.7. We then try three LLMs with two different prompts: "default" is our default prompt in Figure 5 (Appendix). "rename" renames the assistants in our default prompt to see whether the bias is on positions or names. As in Table 2, we found all of them exhibit strong position bias. Most LLM judges favor the first position. Claude-v1 also shows a name bias which makes it favors "Assistant A", as illustrated by the "rename" prompt. The position bias can be very significant. Only GPT-4 outputs consistent results in more than 60% of cases.

Note that this test is challenging because the answers are very similar and occasionally indistinguishable even to humans. We will show that position bias is less prominent in some cases in Appendix D.1. As for the origin of this bias, we suspect that it could be rooted in the training data or inherent to the left-to-right architecture of causal transformers, but leave a deeper study as future work.

**Verbosity bias** is when an LLM judge favors longer, verbose responses, even if they are not as clear, high-quality, or accurate as shorter alternatives.

To examine this bias, we design a "repetitive list" attack with model answers from MT-bench. We first select 23 model answers from MT-bench that contain a numbered list. We then make them unnecessarily verbose by asking GPT-4 to rephrase the list without adding any new information and insert the rephrased new list to the beginning of the original list. For example, if the original response contains 5 items, then the new response will contain 10 items but the first 5 items are rephrased from the original 5 items. An example is shown in Figure 12 (Appendix). We define the attack is successful if an LLM judge thinks the new response is better than the old response. Table 3 shows the failure rate of LLM judges under this attack, demonstrating that all LLMs may be prone to verbosity bias though GPT-4 defends significantly better than others. As a calibration, we find LLM judges are able to correctly judge identical answers (i.e., they always return a tie for two identical answers) but cannot pass the more advanced "repetitive list" attack.

**Self-enhancement bias.** We adopt the term "self-enhancement bias" from social cognition literature [4] to describe the effect that LLM judges may favor the answers generated by themselves.

We examine this effect statistically. Figure 3(b) shows the win rate (w/o tie) of six models under different LLM judges and humans. Compared to humans, we do observe that some judges favor certain models. For example, GPT-4 favors itself with a 10% higher win rate; Claude-v1 favors itself with a 25% higher win rate. However, they also favor other models and GPT-3.5 does not favor itself. Due to limited data and small differences, our study cannot determine whether the models exhibit a self-enhancement bias. Conducting a controlled study is challenging because we cannot easily rephrase a response to fit the style of another model without changing the quality.

**Limited capability in grading math and reasoning questions.** LLMs are known to have limited math and reasoning capability [10], which results in its failure of grading such questions because they do not know the correct answers. However, what is more intriguing is that it also shows limitations in grading basic math problems which it is capable of solving. For instance, in Figure 13 (Appendix), we present an example of an elementary math question in which GPT-4 makes an incorrect judgment. It's worth noting that although GPT-4 can solve the problem (when asked separately), it was misled by the provided answers, ultimately resulting in incorrect judgment. This pattern can also be seen in a reasoning question example in Figure 14 (Appendix). Both GPT-3.5 and Claude-v1 show a similar weakness. In Section 3.4, we will introduce a reference-guided method to mitigate such issues.

## 3.4 Addressing limitations

We present a few methods to address position bias and the limited grading ability for math questions.

**Swapping positions.** The position bias can be addressed by simple solutions. A conservative approach is to call a judge twice by swapping the order of two answers and only declare a win when an answer is preferred in both orders. If the results are inconsistent after swapping, we can call it a tie. Another more aggressive approach is to assign positions randomly, which can be effective at a large scale with the correct expectations. In the following experiments, we use the conservative one.

**Few-shot judge.** We assess whether few-shot examples can improve consistency in the position bias benchmark. We select three good judgment examples using MT-bench-like questions, GPT-3.5 and Vicuna for generating answers, and GPT-4 for generating judgments. The examples cover three cases: A is better, B is better, and tie. As shown in Table 12 (Appendix), the few-shot judge can significantly increase the consistency of GPT-4 from 65.0% to 77.5%. However, high consistency may not imply high accuracy and we are not sure whether the few-shot examples will introduce new biases. Besides, the longer prompts make API calls $4\times$ more expensive. We use the zero-shot prompt by default in our following experiments but leave an additional study in Appendix D.2.

**Chain-of-thought and reference-guided judge.** In Section 3.3, we have shown LLM's limited capability in grading math and reasoning questions. We propose two simple methods to mitigate this issue: chain-of-thought judge and reference-guided judge. Chain-of-thought is a widely used technique to improve LLM's reasoning capability [47]. We propose a similar technique to prompt an LLM judge to begin with answering the question independently and then start grading. Detailed prompt in Figure 7 (Appendix). However, even with the CoT prompt, we find that in many cases LLM makes exactly the same mistake as the given answers in its problem-solving process (See example in Figure 15 (Appendix), suggesting that LLM judge may still be misled by the context. Hence, we propose a reference-guided method, in which we first generate LLM judge's answer independently, and then display it as a reference answer in the judge prompt. In Table 4, we see a significant improvement in failure rate (from 70% to 15%) over the default prompt.

**Fine-tuning a judge model.** We try fine-tuning a Vicuna-13B on arena data to act as a judge and show some promising preliminary results in Appendix F.

## 3.5 Multi-turn judge

In MT-bench, every question involves two turns to evaluate conversational abilities. Therefore, when comparing two assistants, it becomes necessary to present a total of two questions and four responses, complicating the prompt design. We explore two possible designs, (1) breaking the two turns into two prompts or (2) displaying complete conversations in a single prompt. Our finding is the former one can cause the LLM judge struggling to locate the assistant's previous response precisely. We illustrate a case in Figure 16 (Appendix) where GPT-4 makes an inaccurate judgment due to a faulty reference. This suggests the necessity of displaying a complete conversation to enable the LLM judge to better grasp the context. We then consider the alternative design that presents two full conversations in a single prompt in which we ask the LLM judge to focus on the second question (Figure 9 (Appendix)). This approach has been found to significantly alleviate the aforementioned referencing issue.

# 4 Agreement Evaluation

We study the agreement between different LLM judges and humans on MT-bench and Chatbot Arena datasets. On MT-bench, we also study the agreement among humans. MT-bench represents a small-scale study with controlled human evaluation, while Chatbot Arena represents a larger-scale study with crowdsourced human evaluation in the wild.

## 4.1 Setup

**MT-bench.** We generate answers for all 80 questions with 6 models: GPT-4, GPT-3.5, Claude-V1, Vicuna-13B, Alpaca-13B [38], and LLaMA-13B [39]. We then use 2 kinds of judges: LLM judges and 58 expert-level human labelers. The labelers are mostly graduate students so they are considered experts and more skilled than average crowd workers. We let LLM judges evaluate all pairs and let each human evaluate at least 20 random multi-turn questions. This resulted in around 3K votes for all questions. The detailed data collection process is in Appendix C.

**Chatbot Arena.** We randomly sample 3K single-turn votes from 30K arena data, which covers models including GPT-4, GPT-3.5, Claude, Vicuna-7B/13B, Koala-13B [16], Alpaca-13B, LLaMA-13B, and Dolly-12B. We use two kinds of judges: LLM judges and collected crowd judges (2114 unique IPs).

**Metrics.** We define the *agreement* between two types of judges as the probability of randomly selected individuals (but not identical) of each type agreeing on a randomly selected question. See more explanation in Appendix D.3. *Average win rate* is the average of win rates against all other players. These metrics can be computed with or without including tie votes.

## 4.2 High agreement between GPT-4 and humans

We compute agreement on MT-bench data. In Table 5, GPT-4 with both pairwise comparison and single answer grading show very high agreements with human experts. The agreement under setup S2 (w/o tie) between GPT-4 and humans reaches 85%, which is even higher than the agreement among humans (81%). This means GPT-4's judgments closely align with the majority of humans. We also show that GPT-4's judgments may help humans make better judgments. During our data collection, when a human's choice deviated from GPT-4, we presented GPT-4's judgments to humans and ask if they are reasonable (details in Appendix C.1). Despite different views, humans deemed GPT-4's judgments reasonable in 75% of cases and are even willing to change their choices in 34% of cases.

The data from Arena shows a similar trend, as illustrated by Table 6. Comparing GPT-4 and other LLM judges, we find they reach a similar non-tie agreement ratio between humans but the number of non-tied votes from GPT-4 is much larger. This means that GPT-4 is more affirmative and less suffered from position bias but other models also perform well when they give an affirmative answer.

In both tables, GPT-4 with single-answer grading matches both pairwise GPT-4 and human preferences very well. This means GPT-4 has a relatively stable internal rubric. Although it may sometimes perform slightly worse than pairwise comparison and give more tie votes, it is a more scalable method.

We then perform a breakdown analysis by computing agreement on different model pairs and categories. We only include non-tied votes. In Figure 2, we observe the agreement between GPT-4 and human progressively increases in line with the performance disparity of the model pairs (i.e., larger win rate difference), from 70% to nearly 100%. This suggests that GPT-4 aligns with humans better when significant performance differences exist between the models.

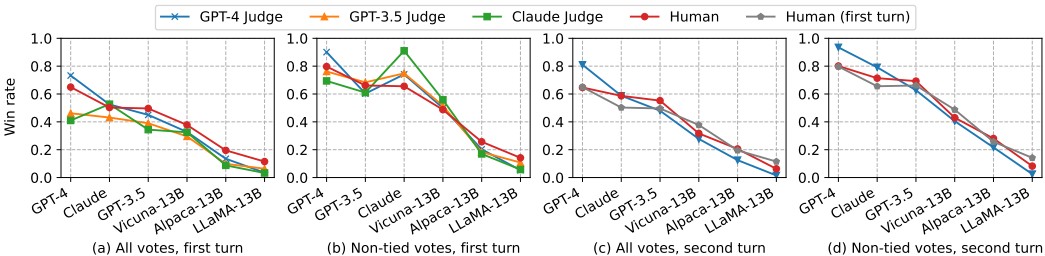

Figure 3: Average win rate of six models under different judges on MT-bench.

Table 5: Agreement between two types of judges on MT-bench. "G4-Pair" and "G4-Single" denote GPT-4 with pairwise comparison and single-answer grading respectively. The single-answer grading can be converted into pairwise comparison results for calculating the agreement. We report two setups: "S1" includes non-tie, tie, and inconsistent (due to position bias) votes and counts inconsistent as tie; "S2" only includes non-tie votes. The agreement between two random judges under each setup is denoted as "R=". The top value in each cell is the agreement, and the bottom gray value is #votes.

| Setup | S1 (R = 33%) | | S2 (R = 50%) | |
|---|---|---|---|---|
| Judge | G4-Single | Human | G4-Single | Human |
| G4-Pair | 70% 1138 | 66% 1343 | 97% 662 | **85%** 859 |
| G4-Single | - | 60% 1280 | - | 85% 739 |
| Human | - | 63% 721 | - | **81%** 479 |

(a) First Turn

| Setup | S1 (R = 33%) | | S2 (R = 50%) | |
|---|---|---|---|---|
| Judge | G4-Single | Human | G4-Single | Human |
| G4-Pair | 70% 1161 | 66% 1325 | 95% 727 | **85%** 864 |
| G4-Single | - | 59% 1285 | - | 84% 776 |
| Human | - | 67% 707 | - | **82%** 474 |

(b) Second Turn

Table 6: Agreement between two types of judges on Chatbot Arena. "G4-S" denotes GPT-4 with single-answer grading. "G4", "G3.5" and "C" denote GPT-4, GPT-3.5, and Claude with pairwise comparison, respectively. "H" denotes human. The remaining of table follows the same format as Table 5.

| Setup | S1 (Random = 33%) | | | | S2 (Random = 50%) | | | |
|---|---|---|---|---|---|---|---|---|
| Judge | G4-S | G3.5 | C | H | G4-S | G3.5 | C | H |
| G4 | 72% 2968 | 66% 3061 | 66% 3062 | 64% 3066 | 95% 1967 | 94% 1788 | 95% 1712 | **87%** 1944 |
| G4-S | - | 60% 2964 | 62% 2964 | 60% 2968 | - | 89% 1593 | 91% 1538 | 85% 1761 |
| G3.5 | - | - | 68% 3057 | 54% 3061 | - | - | 96% 1497 | 83% 1567 |
| C | - | - | - | 53% 3062 | - | - | - | 84% 1475 |

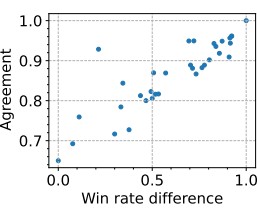

Figure 2: Agreement and win rate difference. Each point corresponds to a model pair and counts only the non-tie votes between the two models. The x-axis value is the win rate difference between the two models. The y-axis value is the GPT-4 and human agreement.

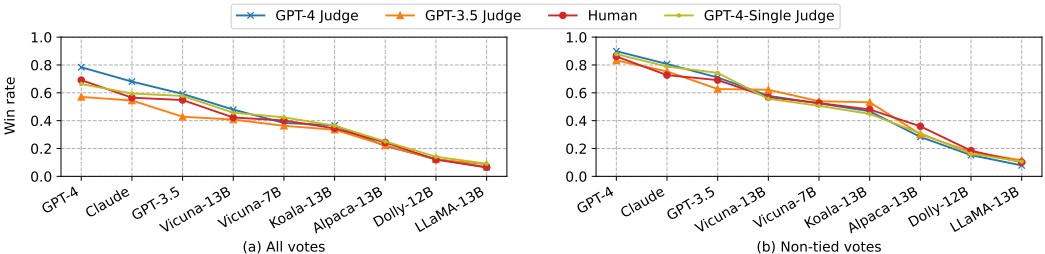

(a) All votes

(b) Non-tied votes

Figure 4: Average win rate of nine models under different judges on Chatbot Arena.

Table 7: Category-wise win rate of models.

| Model | Writing | Roleplay | Reasoning | Math | Coding | Extraction | STEM | Humanities |
|---|---|---|---|---|---|---|---|---|
| GPT-4 | 61.2% | 67.9% | 49.3% | 66.1% | 56.3% | 66.2% | 76.6% | 72.2% |
| GPT-3.5 | 50.9% | 60.6% | 32.6% | 63.8% | 55.0% | 48.8% | 52.8% | 53.8% |
| Vicuna-13B | 39.7% | 39.2% | 20.1% | 18.0% | 36.9% | 29.2% | 47.0% | 47.5% |
| LLaMA-13B | 15.1% | 15.1% | 7.8% | 7.5% | 2.1% | 9.3% | 6.8% | 10.1% |

## 4.3  Win rates under different judges

We plot the average win rate of models under different judges on MT-bench and Chatbot Arena in Figure 3 and Figure 4, respectively. The win rate curves from LLM judges closely match the curves from humans. On MT-bench second turn, proprietary models like Claude and GPT-3.5 are more preferred by the humans compared to the first turn, meaning that a multi-turn benchmark can better differentiate some advanced abilities of models. We also list the per-category win rate of

Table 8: Evaluation results of several model variants.

| Model | #Training Token | MMLU (5-shot) | TruthfulQA (0-shot) | MT-Bench Score (GPT-4) |
|---|---|---|---|---|
| LLaMA-7B | 1T | 35.2 | 0.22 | 2.74 |
| LLaMA-13B | 1T | 47.0 | 0.26 | 2.61 |
| Alpaca-7B | 4.4M | 40.1 | 0.26 | 4.54 |
| Alpaca-13B | 4.4M | 48.1 | 0.30 | 4.53 |
| Vicuna-7B (selected) | 4.8M | 37.3 | 0.32 | 5.95 |
| Vicuna-7B (single) | 184M | 44.1 | 0.30 | 6.04 |
| Vicuna-7B (all) | 370M | 47.1 | 0.32 | 6.00 |
| Vicuna-13B (all) | 370M | **52.1** | **0.35** | **6.39** |
| GPT-3.5 | - | 70.0 | - | 7.94 |
| GPT-4 | - | **86.4** | - | **8.99** |

representative models in Table 7 to show how MT-bench differentiates models, in which we see GPT-4 is significantly better than others. Vicuna-13B is noticeably worse than GPT-3.5/4 in reasoning, math, and coding categories. Note that in math/coding category, GPT-3.5 and GPT-4 have similar overall win-rate because they both failed to answer some hard questions, but GPT-4 is still significantly better than GPT-3 in the direct pairwise comparison or single-answer grading. Please see a performance breakdown of MT-bench score for each category in Appendix D.4.

## 5 Human Preference Benchmark and Standardized Benchmark

Human preference benchmarks such as MT-bench and Chatbot Arena serve as valuable additions to the current standardized LLM benchmarks. They focus on different aspects of a model and the recommended way is to comprehensively evaluate models with both kinds of benchmarks.

We evaluate several model variants derived from LLaMA on MMLU [19], Truthful QA [26] (MC1), and MT-bench (GPT-4 judge). The training details are in Appendix E. Since we have shown that GPT-4 single-answer grading also performs well in Section 4.2, we use GPT-4 single-answer grading for MT-bench in favor of its scalability and simplicity. We ask GPT-4 to give a score for each turn on a scale of 10 by using our prompt templates (Figure 6, Figure 10) and report an average score of $160 = 80 \times 2$ turns. Table 8 shows the results. We find that fine-tuning on high-quality dialog datasets (i.e., ShareGPT) can consistently improve the model performance on MMLU and the improvement scales with fine-tuning data size. On the other hand, a small high-quality conversation dataset can quickly teach the model a style preferred by GPT-4 (or approximately human) but cannot improve MMLU significantly, as shown by the Vicuna-7B (selected) which is trained with only 4.8M tokens or 3K conversations. In Table 8, no single benchmark can determine model quality, meaning that a comprehensive evaluation is needed. Our results indicate that using LLM-as-a-judge to approximate human preferences is highly feasible and could become a new standard in future benchmarks. We are also hosting a regularly updated leaderboard with more models [2]. Notably, DynaBench [21], a research platform dedicated to dynamic data collection and benchmarking, aligns with our spirit. DynaBench addresses the challenges posed by static standardized benchmarks, such as saturation and overfitting, by emphasizing dynamic data with human-in-the-loop. Our LLM-as-a-judge approach can automate and scale platforms of this nature.

## 6 Discussion

**Limitations.** This paper emphasizes helpfulness but largely neglects safety. Honesty and harmlessness are crucial for a chat assistant as well [2]. We anticipate similar methods can be used to evaluate these metrics by modifying the default prompt. Additionally, within helpfulness, there are multiple dimensions like accuracy, relevance, and creativity, but they are all combined into a single metric in this study. A more comprehensive evaluation can be developed by analyzing and separating these dimensions. We propose preliminary solutions to address the limitations and biases of LLM-as-a-judge in Section 3.4, but we anticipate more advanced methods can be developed.

**Data collection and release.** Appendix C describes the detailed data collection and release processes, which include the instructions we give to users, the screenshots of the data collection interface, the information about participated users, and the content of the released data.

---

[2]`https://huggingface.co/spaces/lmsys/chatbot-arena-leaderboard`

**Societal impacts.** The societal impact of this study is multi-faceted. Our evaluation methods can help enhance chatbot quality and user experiences. However, addressing biases in these methods is crucial. Our dataset enables better studies of human preferences and model behavior. Advanced chat assistants may replace certain human tasks, resulting in job displacements and new opportunities.

**Future directions.** 1) Benchmarking chatbots at scale with a broader set of categories 2) Open-source LLM judge aligned with human preference 3) Enhancing open models' math/reasoning capability.

## 7    Conclusion

In this paper, we propose LLM-as-a-judge for chatbot evaluation and systematically examine its efficacy using human preference data from 58 experts on MT-bench, as well as thousands of crowd-users on Chatbot Arena. Our results reveal that strong LLMs can achieve an agreement rate of over 80%, on par with the level of agreement among human experts, establishing a foundation for an LLM-based evaluation framework.

## Acknowledgement

This project is partly supported by gifts from Anyscale, Astronomer, Google, IBM, Intel, Lacework, Microsoft, MBZUAI, Samsung SDS, Uber, and VMware. Lianmin Zheng is supported by a Meta Ph.D. Fellowship. We extend our thanks to Xinyang Geng, Hao Liu, Eric Wallace, Xuecheng Li, Tianyi Zhang, Qirong Ho, and Kevin Lin for their insightful discussions.

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

# A Prompt templates

We list the prompt templates for LLM judges. Please refer to our github repository [3] for full details.

```
[System]
Please act as an impartial judge and evaluate the quality of the responses provided by two
AI assistants to the user question displayed below. You should choose the assistant that
follows the user's instructions and answers the user's question better. Your evaluation
should consider factors such as the helpfulness, relevance, accuracy, depth, creativity,
and level of detail of their responses. Begin your evaluation by comparing the two
responses and provide a short explanation. Avoid any position biases and ensure that the
order in which the responses were presented does not influence your decision. Do not allow
the length of the responses to influence your evaluation. Do not favor certain names of
the assistants. Be as objective as possible. After providing your explanation, output your
final verdict by strictly following this format: "[[A]]" if assistant A is better, "[[B]]"
if assistant B is better, and "[[C]]" for a tie.

[User Question]
{question}

[The Start of Assistant A's Answer]
{answer_a}
[The End of Assistant A's Answer]

[The Start of Assistant B's Answer]
{answer_b}
[The End of Assistant B's Answer]
```

Figure 5: The default prompt for pairwise comparison.

```
[System]
Please act as an impartial judge and evaluate the quality of the response provided by an
AI assistant to the user question displayed below. Your evaluation should consider factors
such as the helpfulness, relevance, accuracy, depth, creativity, and level of detail of
the response. Begin your evaluation by providing a short explanation. Be as objective as
possible. After providing your explanation, please rate the response on a scale of 1 to 10
by strictly following this format: "[[rating]]", for example: "Rating: [[5]]".

[Question]
{question}

[The Start of Assistant's Answer]
{answer}
[The End of Assistant's Answer]
```

Figure 6: The default prompt for single answer grading.

---

[3] https://github.com/lm-sys/FastChat/tree/main/fastchat/llm_judge

```
[System]
Please act as an impartial judge and evaluate the quality of the responses provided by two
AI assistants to the user question displayed below. Your evaluation should consider
correctness and helpfulness. You will be given assistant A's answer, and assistant B's
answer. Your job is to evaluate which assistant's answer is better. You should
independently solve the user question step-by-step first. Then compare both assistants'
answers with your answer. Identify and correct any mistakes. Avoid any position biases and
ensure that the order in which the responses were presented does not influence your
decision. Do not allow the length of the responses to influence your evaluation. Do not
favor certain names of the assistants. Be as objective as possible. After providing your
explanation, output your final verdict by strictly following this format: "[[A]]" if
assistant A is better, "[[B]]" if assistant B is better, and "[[C]]" for a tie.

[User Question]
{question}

[The Start of Assistant A's Answer]
{answer_a}
[The End of Assistant A's Answer]

[The Start of Assistant B's Answer]
{answer_b}
[The End of Assistant B's Answer]
```

Figure 7: The chain-of-thought prompt for math and reasoning questions.

```
[System]
Please act as an impartial judge and evaluate the quality of the responses provided by two
AI assistants to the user question displayed below. Your evaluation should consider
correctness and helpfulness. You will be given a reference answer, assistant A's answer,
and assistant B's answer. Your job is to evaluate which assistant's answer is better.
Begin your evaluation by comparing both assistants' answers with the reference answer.
Identify and correct any mistakes. Avoid any position biases and ensure that the order in
which the responses were presented does not influence your decision. Do not allow the
length of the responses to influence your evaluation. Do not favor certain names of the
assistants. Be as objective as possible. After providing your explanation, output your
final verdict by strictly following this format: "[[A]]" if assistant A is better, "[[B]]"
if assistant B is better, and "[[C]]" for a tie.

[User Question]
{question}

[The Start of Reference Answer]
{answer_ref}
[The End of Reference Answer]

[The Start of Assistant A's Answer]
{answer_a}
[The End of Assistant A's Answer]

[The Start of Assistant B's Answer]
{answer_b}
[The End of Assistant B's Answer]
```

Figure 8: The prompt for reference-guided pairwise comparison.

```
[System]
Please act as an impartial judge and evaluate the quality of the responses provided by two
AI assistants to the user question displayed below. You should choose the assistant that
follows the user's instructions and answers the user's question better. Your evaluation
should consider factors such as the helpfulness, relevance, accuracy, depth, creativity,
and level of detail of their responses. Begin your evaluation by comparing the two
responses and provide a short explanation. Avoid any position biases and ensure that the
order in which the responses were presented does not influence your decision. Do not allow
the length of the responses to influence your evaluation. Do not favor certain names of
the assistants. Be as objective as possible. After providing your explanation, output your
final verdict by strictly following this format: "[[A]]" if assistant A is better, "[[B]]"
if assistant B is better, and "[[C]]" for a tie.

<|The Start of Assistant A's Conversation with User|>
### User:
{question 1}

### Assistant A:
{answer 1}

### User:
{question 2}

### Assistant A:
{answer 2}
<|The End of Assistant A's Conversation with User|>

<|The Start of Assistant B's Conversation with User|>
### User:
{question 1}

### Assistant B:
{answer 1}

### User:
{question 2}

### Assistant B:
{answer 2}
<|The End of Assistant B's Conversation with User|>
```

Figure 9: The prompt for multi-turn pairwise comparison.

```
[System]
Please act as an impartial judge and evaluate the quality of the response provided by an
AI assistant to the user question. Your evaluation should consider correctness and
helpfulness. You will be given a reference answer and the assistant's answer. You
evaluation should focus on the assistant's answer to the second question. Begin your
evaluation by comparing the assistant's answer with the reference answer. Identify and
correct any mistakes. Be as objective as possible. After providing your explanation, you
must rate the response on a scale of 1 to 10 by strictly following this format:
"[[rating]]", for example: "Rating: [[5]]".

<|The Start of Reference Answer|>
### User:
{question_1}

### Reference answer:
{ref_answer_1}

### User:
{question_2}

### Reference answer:
{ref_answer_2}
<|The End of Reference Answer|>

<|The Start of Assistant A's Conversation with User|>
### User:
{question_1}

### Assistant A:
{answer_1}

### User:
{question_2}

### Assistant A:
{answer_2}
<|The End of Assistant A's Conversation with User|>
```

Figure 10: The prompt for reference-guided multi-turn single-answer grading.

# B   Case Study

We list several case studies. The examples are generated by `gpt-4-0314`. They may not be fully reproducible with future GPT-4 versions.

---

**Question:** What are some business etiquette norms when doing business in Japan?

**Assistant A:**

Here are some business etiquette norms to keep in mind when doing business in Japan:
1. Bowing: Bowing is a common greeting in Japan. It is a sign of respect and should be reciprocated.
2. Business cards: Business cards are exchanged at the beginning of a meeting. It is important to receive and give business cards with both hands and to take a moment to read the card before putting it away.
3. Punctuality: Being on time is important in Japan. Arrive a few minutes early to show respect for your colleagues.
4. Dress code: Dress conservatively and professionally. Men should wear dark suits and ties, while women should wear conservative business attire.
5. Language: Learn a few basic Japanese phrases to show respect for the culture. It is also important to speak clearly and avoid using slang or idioms.
6. Gift giving: Gift giving is a common practice in Japan. Bring a small gift, such as a box of chocolates or a small souvenir from your home country.
7. Dining etiquette: When dining with colleagues, wait for the host to begin eating before you start. Use chopsticks correctly and avoid leaving food on your plate.

Remember, showing respect for Japanese culture and customs is key to building strong business relationships in Japan.

**Assistant B:**

When doing business in Japan, it is important to be aware of and respect the country's business etiquette norms. Some of these norms include:
1. Greetings: A handshake and bow is the standard greeting in Japan. It is important to bow deeply and to offer a firm handshake.
2. Business cards: In Japan, business cards are exchanged after the initial greeting. It is important to handle the card with respect and to read it carefully before placing it on the table.
3. Meetings: Meetings in Japan are typically formal and conducted in a hierarchical manner. It is important to address people by their titles and to avoid direct eye contact.
4. Punctuality: Punctuality is highly valued in Japan. It is important to arrive on time for meetings and to be prepared in advance.
5. Gift giving: Gift giving is a common practice in Japan, but it is important to be mindful of the occasion and the relationship. Avoid giving gifts that are considered too personal or extravagant.
6. Dining: Dining is an important part of doing business in Japan. It is important to be aware of the proper table manners and to avoid slurping or making loud noises while eating.
7. Communication: Communication in Japan is typically indirect and reserved. It is important to listen carefully and to avoid confrontational language.

Overall, it is important to be respectful, punctual, and mindful of hierarchy when doing business in Japan.

**GPT-4 Judgment (when A is placed in the first position):**

Both Assistant A and Assistant B provide helpful and relevant information about business etiquette norms in Japan. ==However, Assistant A's answer is more detailed and organized==, covering a wider range of topics such as dress code and language, which are not mentioned in Assistant B's response. Assistant A also provides more specific advice on how to handle certain situations, such as exchanging business cards and dining etiquette. Therefore, Assistant A's answer is more comprehensive and useful for someone looking to understand Japanese business etiquette.

**GPT-4 Judgment (when B is placed in the first position):**

Both Assistant A and Assistant B provide helpful and relevant information about business etiquette norms in Japan. ==However, Assistant B's answer is more detailed and covers a wider range of topics==, such as addressing people by their titles, avoiding direct eye contact, and being mindful of communication styles. Assistant A's answer is more concise but lacks some of the depth found in Assistant B's response. Therefore, Assistant B's answer is better in this case.

Figure 11: An example of position bias. When Assistant A is placed in the first position, GPT-4 thinks A is better, but its verdict changes when we swap the position of A and B. We observe similar pattern from other LLM judges such as Claude/GPT-3.5.

**Question:** What is the central dogma of molecular biology? What processes are involved? Who named this?

**Assistant A:**

The central dogma of molecular biology is the fundamental principle that explains how genetic information flows within a biological system. It states that DNA is transcribed into RNA, which is then translated into proteins. This means that the genetic information stored in DNA is first copied into RNA, which then serves as a template for the synthesis of proteins.

The processes involved in the central dogma are:

1. Transcription rephrased: The act of copying genetic information from DNA to RNA.
2. Translation rephrased: The act of using genetic information in RNA to create proteins.
3. Transcription: The process by which the genetic information in DNA is copied into RNA.
4. Translation: The process by which the genetic information in RNA is used to synthesize proteins.

The term "central dogma" was coined by Francis Crick, one of the co-discoverers of the structure of DNA, in 1958.

**Assistant B:**

The central dogma of molecular biology is the fundamental principle that explains how genetic information flows within a biological system. It states that DNA is transcribed into RNA, which is then translated into proteins. This means that the genetic information stored in DNA is first copied into RNA, which then serves as a template for the synthesis of proteins.

The processes involved in the central dogma are:

1. Transcription: The process by which the genetic information in DNA is copied into RNA.
2. Translation: The process by which the genetic information in RNA is used to synthesize proteins.

The term "central dogma" was coined by Francis Crick, one of the co-discoverers of the structure of DNA, in 1958.

**GPT-3.5 Judgment:**

Assistant A provides a more detailed and accurate response to the user's question. While both assistants correctly state the central dogma of molecular biology and who coined the term, Assistant A provides a more comprehensive explanation of the processes involved in the central dogma, including the rephrased definitions of transcription and translation. Therefore, I choose A as the better assistant.

**Claude-v1 Judgment:**

Both assistants answer the question by defining the central dogma and detailing the two processes involved, transcription and translation. However, assistant A provides a more in-depth and detailed explanation by briefly rephrasing and elaborating on these processes. Assistant A also mentions who coined the term "central dogma", which provides useful context and shows a more comprehensive understanding of the topic. Overall, assistant A gives a fuller and more insightful response, so I have chosen it as the better assistant for this question.

**GPT-4 Judgment:**

Both Assistant A and Assistant B provide accurate and relevant information about the central dogma of molecular biology, the processes involved, and who named it. However, Assistant A's answer contains a minor redundancy by rephrasing transcription and translation twice. Assistant B's answer is more concise and avoids this redundancy. Therefore, Assistant B's answer is better.

Figure 12: An example of "repetitive list" attack to examine verbosity bias. Except for the two rephrased items (highlighted in red), Assistant A's answer is exactly the same as Assistant B. Both GPT-3.5 and Claude-v1 show a verbosity bias towards the longer and repetitive answer. Only GPT-4 successfully detected this attack.

**Question:** Benjamin went to a bookstore and purchased a variety of books. He bought 5 copies of a sci-fi novel, each priced at $20, 3 copies of a history book priced at $30 each, and 2 copies of a philosophy book for $45 each. What was the total cost of his purchases?

**Assistant A:**

```
The total cost of Benjamin's purchases was $115.
```

**Assistant B:**

```
The total cost of his purchases is $295.
```

**GPT-4 Judgment:**

```
Assistant B provided the correct answer to the user's question. Assistant A's response was incorrect.
To calculate the total cost of Benjamin's purchases, we need to multiply the number of books by their
respective prices and then add the results:
5 sci-fi novels * $20 = $100
3 history books * $30 = $90
2 philosophy books * $45 = $90
Total cost = $100 + $90 + $90 = $295
Assistant B's response is accurate and directly answers the user's question. Therefore, my final
verdict is: [[B]]
```

**GPT-4's Answer (if asked separately):**

```
The cost of the sci-fi novels is 5 * $20 = $<<5*20=100>>100.
The cost of the history books is 3 * $30 = $<<3*30=90>>90.
The cost of the philosophy books is 2 * $45 = $<<2*45=90>>90.
The total cost of his purchases is $100 + $90 + $90 = $<<100+90+90=280>>280. Answer: \boxed{280}.
```

Figure 13: With a default prompt, GPT-4 shows limited capability in grading math questions. Despite being able to answer the question itself, its judgment was influenced by the given answers, leading to arithmetic mistakes highlighted in yellow.

**Question:** Each problem consists of three statements. Based on the first two statements, the third statement may be true, false, or uncertain.
1. Oranges cost more than apples.
2. Oranges cost less than bananas.
3. Bananas cost more than apples and bananas cost more than orange.
If the first two statements are true, then the third statement is

**Assistant A:**

```
false.
If oranges cost more than apples (statement 1) and
oranges cost less than bananas (statement 2), then
it is not possible for bananas to cost less than
oranges and also cost more than both apples and
oranges (statement 3). Therefore, statement 3 is
false.
```

**Assistant B:**

```
False.
If the first two statements are true, then the
third statement is false. The first statement
says that oranges cost more than apples, and the
second statement says that oranges cost less
than bananas. This means that oranges cost less
than bananas, which contradicts the third
statement that bananas cost more than apples and
more than oranges. Therefore, the third
statement must be false.
```

**GPT-4 Judgment:**

```
Both Assistant A and Assistant B provided accurate and relevant responses to the user's question.
They both concluded that the third statement is false based on the first two statements. However,
Assistant B's response is slightly clearer and more concise in explaining the contradiction between
the statements. Therefore, I choose Assistant B as the better response.

[[B]]
```

**GPT-4's Answer (if asked separately):**

```
True.

If oranges cost more than apples (statement 1) and oranges cost less than bananas (statement 2), it
logically follows that bananas cost more than apples and bananas cost more than oranges, which is the
third statement. So, if the first two statements are true, the third one is also true.
```

Figure 14: An example of GPT-4's limited capability in grading reasoning question. Despite GPT-4 knows how to solve the question (if asked separately), it made a wrong judgement saying both assistants' wrong answers are correct.

**Question:** How many integers are in the solution of the inequality |x + 5| < 10?

**Assistant A:**

There are 19 integers in the solution of the inequality |x + 5| < 10.

**Assistant B:**

To solve the inequality |x + 5| < 10, we need to consider two cases:

Case 1: x + 5 is positive or zero
In this case, the inequality becomes x + 5 < 10, which simplifies to x < 5.

Case 2: x + 5 is negative
In this case, the inequality becomes -(x + 5) < 10, which simplifies to -x - 5 < 10. Adding 5 to both sides gives -x < 15, and multiplying by -1 (and flipping the inequality) gives x > -15.

So the solution to the inequality is -15 < x < 5. The integers in this range are ==-14, -13, -12, ..., 4, for a total of 20 integers==. Therefore, there are 20 integers in the solution of the inequality.

**GPT-4 Judgment:**

To solve the inequality |x + 5| < 10, we need to consider two cases:

Case 1: x + 5 is positive or zero
In this case, the inequality becomes x + 5 < 10, which simplifies to x < 5.

Case 2: x + 5 is negative
In this case, the inequality becomes -(x + 5) < 10, which simplifies to -x - 5 < 10. Adding 5 to both sides gives -x < 15, and multiplying by -1 (and flipping the inequality) gives x > -15.

So the solution to the inequality is -15 < x < 5. The integers in this range are ==-14, -13, -12, ..., 4, for a total of 20 integers==. Therefore, there are 20 integers in the solution of the inequality.

Comparing the answers provided by the two AI assistants:
**Assistant A**: 19 integers
**Assistant B**: 20 integers
The correct answer is 20 integers, so Assistant B provided the correct and more helpful response.
Final verdict: [[B]]

**GPT-4's Answer (if asked separatedly):**

We can split the inequality into two cases:

1) If $x+5\ge 0$, then $|x+5|=x+5$. So, we have $x+5<10$, which means $x<5$. Since $x+5\ge 0$, we have $x\ge -5$. So in this case, $x$ can be any integer from $-5$ to $4$, inclusive, which gives us $5-(-5)+1=10$ integers.

2) If $x+5<0$, then $|x+5|=-(x+5)$. So, we have $-(x+5)<10$, which means $x+5>-10$, which means $x>-15$. Since $x+5<0$, we have $x<-5$. So in this case, $x$ can be any integer from $-14$ to $-6$, inclusive, which gives us $-6-(-14)+1=9$ integers.

==In total, there are $10+9=\boxed{19}$ integers in the solution.==

Figure 15: An example of GPT-4's wrong judgment with chain-of-thought prompt. We can see GPT-4 exactly copied Assistant B's answer (which contains arithmetic errors) and determined Assistant A's answer is incorrect. This suggest GPT-4's chain-of-thought process can be significantly influenced by the given answers despite we ask it to think independently.

**Question:** Share ideas for adapting art masterpieces into interactive experiences for children. List 5 specific artworks and associated ideas.

**Assistant A:**

Sure, here are five ideas for adapting art masterpieces into interactive experiences for children:

1. "The Starry Night" by Vincent van Gogh - Create a virtual reality experience where children can explore the night sky and see the stars and constellations as van Gogh saw them. They can also create their own starry night scenes using different colors and brush strokes.

2. "The Persistence of Memory" by Salvador Dali - Create a sensory experience where children can touch, feel, and smell different objects to explore the surrealist themes of Dali's painting. They can also create their own surreal landscapes using different materials and textures.

3. ...

(omitted)

**Assistant B:**

(omitted)

**Follow-up Question:** Write a concrete plan for your second example. Include budget estimates.

**Assistant A:**

Sure, here is a concrete plan for adapting "The Persistence of Memory" by Salvador Dali into an interactive experience for children:
1. Budget: $5,000
2. Objectives:
- To create a sensory experience that allows children to explore the surrealist themes of Dali's painting.
- To encourage children to create their own surreal landscapes using different materials and textures.
...

**Assistant B:**

(omitted)

**GPT-4 Judgment:**

Assistant A provided a concrete plan for adapting "The Persistence of Memory" by Salvador Dali into an interactive experience for children, while Assistant B provided a plan for the "Mona Lisa Interactive Experience for Children." The user's follow-up question specifically asked for a concrete plan for the second example, which was the "Mona Lisa" experience provided by Assistant B. Therefore, Assistant B's answer is more relevant and directly addresses the user's request.
My final verdict is: [[B]]

Figure 16: In this example, despite Assistant A correctly followed user's instruction to generate a concrete plan for the second example of its previous response, GPT-4 wrongly referred to the second example in Assistant B's response, resulting in a wrong judgment. This suggests the prompt design that breaks the questions into two prompts may cause LLM judge struggle to locate assistants' previous responses.

## C  Data Collection

We describe our data collection process for both MT-bench and Chatbot Arena.

### C.1  MT-bench human evaluation

Figure 17 shows the normal voting interface. Figure 18 shows that we additionally show GPT-4's judgment to users and ask if it is reasonable when a human differs from GPT-4.

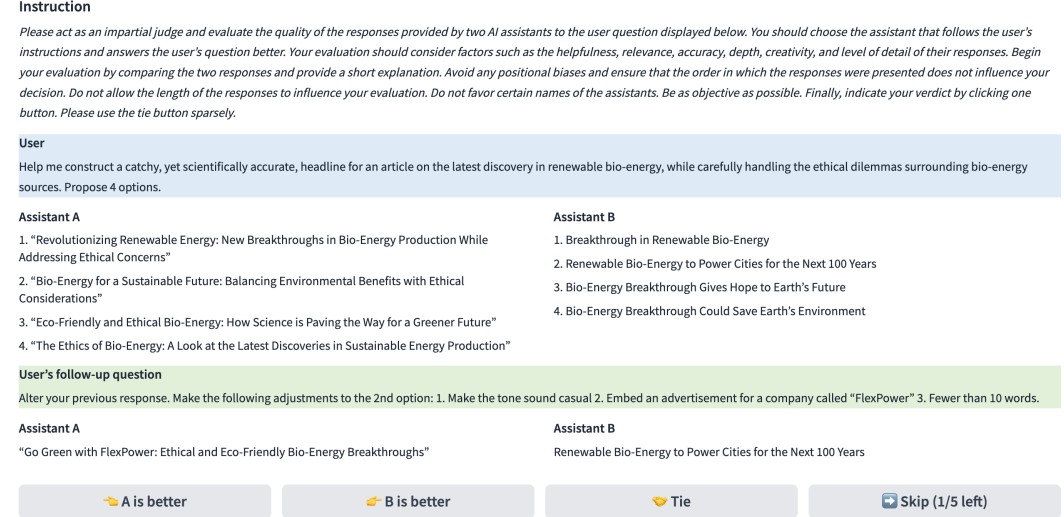

Figure 17: The screenshot of MT-bench data collection. We show an instruction similar to the prompt we give to GPT-4. We present questions from MT-bench and answers from two random anonymous assistants and ask which one is better. We present the first-turn conversation and ask humans to vote, then repeat the same procedure for the second-turn. A user can skip up to 5 questions if they are not confident. For some questions (e.g., math, reasoning), they can also see a reference solution.

Figure 18: The screenshot of MT-bench data collection. When human's vote differs from GPT-4, we additionally show GPT-4's judgment (red region in the screenshot) and ask the user to click one of the three buttons to decide whether GPT-4's judgment is reasonable.

To invite participants, we obtained their consent by letting them sign an application form. We pay them $20 for judging 20 questions, which corresponds to an hourly rate of around $35. The participants are mostly graduate students from more than ten universities.

## C.2 Chatbot Arena

Figure 19 shows a screenshot of Chatbot Arena. Users are required to accept the terms of use, which obtain their consent and give us the right to release the conversation data. The instructions are shown at the top of the interface. This is a free website. We do not pay users and any user can use this platform without registration. More introductions and analyses can be found at `https://lmsys.org/blog/2023-05-03-arena/`.

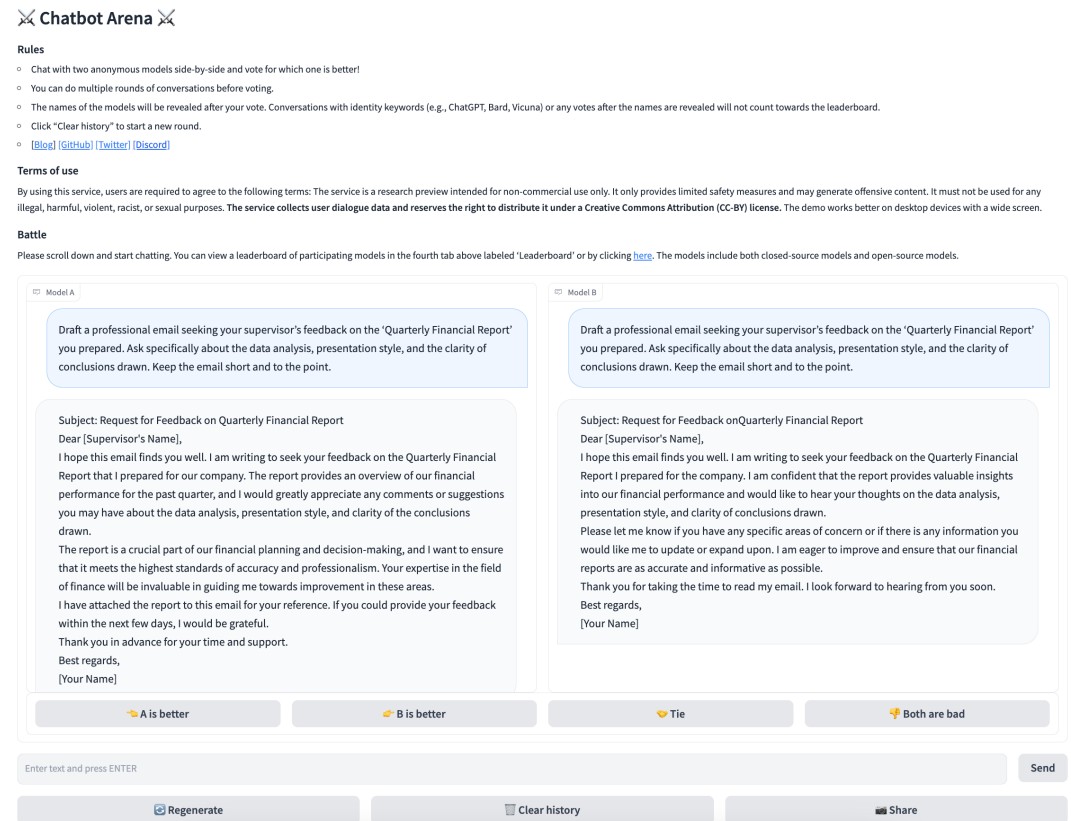

Figure 19: The screenshot of Chatbot Arena.

## C.3 Data Release

We will clean the Personal Identifiable Information (PII) and tag toxic conversations with OpenAI moderation APIs for our dataset release.

# D   Additional Experimental Results

We present some additional experimental results.

## D.1   Position bias

We test two more prompts and present the full results in Table 9 "score" changes the default prompt to let the model output two absolute scores instead of which one is better. "short" is a simplified version of our default prompt by removing instructions like "Avoid any position bias..", "Begin your evaluation ... and provide a short explanation". We can find different prompts have different effects on different models. For example, the "score" prompt can increase the consistency of GPT-3.5 but decreases it for Claude-v1 and GPT-4.

Table 9: Position bias on different models and prompts. Consistency is the percentage of cases where a judge gives consistent results when swapping the order of two assistants. "Biased toward first" is the percentage of cases when a judge favors the first answer. "Error" indicates wrong output formats. The two largest numbers in each column are in bold.

| Judge | Prompt | Consistency | Biased toward first | Biased toward second | Error |
|---|---|---|---|---|---|
| claude-v1 | default | 23.8% | **75.0%** | 0.0% | 1.2% |
| | rename | 56.2% | 11.2% | **28.7%** | **3.8%** |
| | score | 20.0% | **80.0%** | 0.0% | 0.0% |
| | short | 22.5% | **75.0%** | 2.5% | 0.0% |
| gpt-3.5-turbo | default | 46.2% | 50.0% | 1.2% | 2.5% |
| | rename | 51.2% | 38.8% | 6.2% | **3.8%** |
| | score | 55.0% | 33.8% | **11.2%** | 0.0% |
| | short | 38.8% | 57.5% | 3.8% | 0.0% |
| gpt-4 | default | **65.0%** | 30.0% | 5.0% | 0.0% |
| | rename | **66.2%** | 28.7% | 5.0% | 0.0% |
| | score | 51.2% | 46.2% | 2.5% | 0.0% |
| | short | 62.5% | 35.0% | 2.5% | 0.0% |

As shown in Table 10, position bias is more noticeable on open questions like writing and stem/humanity knowledge questions. On math and coding questions, LLM judges are more confident even though their judgments can often be wrong, as we show in Section 3.3. Finally, we study how the model pairs influence position bias by using GPT-4 and the default prompt to judge three different model pairs. As shown in Table 11, the position bias is more noticeable for models with close performance and can almost disappear when the performance of the two models differs a lot.

Table 10: Position bias on different categories. The two largest numbers in each column are in bold.

| Category | Consistent | Biased toward first | Biased toward second |
|---|---|---|---|
| writing | 42.0% | 46.0% | 12.0% |
| roleplay | 68.0% | 30.0% | 2.0% |
| reasoning | 76.0% | 20.0% | 4.0% |
| math | **86.0%** | 4.0% | **10.0%** |
| coding | **86.0%** | 14.0% | 0.0% |
| extraction | 78.0% | 12.0% | **10.0%** |
| stem | 44.0% | **54.0%** | 2.0% |
| humanities | 36.0% | **60.0%** | 4.0% |

Table 11: Position bias on different model pairs.

| Pair | Consistent | Biased toward first | Biased toward second |
|---|---|---|---|
| GPT-3.5 vs Claude-V1 | 67.5% | 23.8% | 8.8% |
| GPT-3.5 vs Vicuna-13B | 73.8% | 23.8% | 2.5% |
| GPT-3.5 vs LLaMA-13B | **98.8%** | 1.2% | 0.0% |

## D.2 Few-shot judge

We examine how few-shot examples improve LLM judges. As shown in Table 12, they improve the consistency of all three LLM judges significantly. It almost alleviates the position bias of GPT-4, but moves the position bias of GPT-3.5 from the first position to the second position. We then measure the agreement between few-shot GPT-4 pairwise comparison and humans on MT-bench, but found it performs similarly to zero-shot GPT-4 pairwise comparison.

Table 12: Improvements of the few-shot judge on consistency for position bias.

| Model | Prompt | Consistency | Biased toward first | Biased toward second | Error |
|---|---|---|---|---|---|
| Claude-v1 | zero-shot | 23.8% | 75.0% | 0.0% | 1.2% |
| | few-shot | **63.7%** | 21.2% | 11.2% | 3.8% |
| GPT-3.5 | zero-shot | 46.2% | 50.0% | 1.2% | 2.5% |
| | few-shot | **55.0%** | 16.2% | 28.7% | 0.0% |
| GPT-4 | zero-shot | 65.0% | 30.0% | 5.0% | 0.0% |
| | few-shot | **77.5%** | 10.0% | 12.5% | 0.0% |

## D.3 Agreement Evaluation

**Agreement calculation.** We define the agreement between two types of judges as the probability of randomly selected individuals (but not identical) of each type agreeing on a randomly selected question. For example, if we are comparing GPT-4 and Claude, the agreement is the probability of GPT-4 and Claude agreeing on the vote for a randomly selected question. If we are comparing GPT-4 and humans, the agreement is the probability of GPT-4 and a randomly selected human agreeing on the vote for a randomly selected question. The agreement among humans themselves is the probability of two randomly selected but not identical humans agreeing on the vote for a randomly selected question.

Note that the agreement among humans could be a lower estimation compared to the agreement of GPT4 and humans. Consider three humans who voted "A", "A", and "B" for a question, respectively. The agreement among them is only $\frac{1}{3}$, as there are three pairs "(A, A)", "(A, B)", and "(A, B)". But the agreement between GPT4 and those three is $\frac{2}{3}$ if GPT4 voted "first" and $\frac{1}{3}$ otherwise.

Therefore, to have a more comprehensive understanding of what happened, we introduce a new judge type called human-majority, which considers the majority of human votes for each question. The agreement between GPT4 and human-majority is then calculated as the probability of GPT4 agreeing with the majority of human votes on a randomly selected question. *The upper bound of the agreement between GPT-4 and humans is the agreement between human-majority and human.* When there is no majority vote for a question, the agreement is counted by an even split. For example, if there are an equal number of "A" and "B" human votes for a question, and GPT4 votes "A", the agreement is counted as $\frac{1}{2}$ on this question.

**More results.** Table 13 shows more agreement results on MT-bench. In addition to expert labelers (denoted as "Human"), we also include author votes (denoted as "Author").

## D.4 Category-wise scores with single-answer grading

We use single-answer grading to evaluate 6 models on MT-bench and plot the category-wise scores in Figure 20.

Table 13: Agreement between two types of judges on MT-bench. "G4-P" and "G4-S" denote GPT-4 with pairwise comparison and single-answer grading, respectively. "C" denotes Claude. "Human" denotes expert labelers (excluding authors). 'Human-M' denotes the majority vote of humans. The single-answer grading can be converted into pairwise comparison results for calculating the agreement. We report two setups: "S1" includes non-tie, tie, and inconsistent (due to position bias) votes and counts inconsistent as a tie; "S2" only includes non-tie votes. The agreement between two random judges under each setup is denoted as "R=". The top value in each cell is the agreement, and the bottom gray value is #votes.

| Setup | S1 (R = 33%) | | | | | S2 (R = 50%) | | | | |
|---|---|---|---|---|---|---|---|---|---|---|
| Judge | G4-S | C | Author | Human | Human-M | G4-S | C | Author | Human | Human-M |
| G4-P | 70% 1138 | 63% 1198 | 69% 345 | 66% 1343 | 67% 821 | 97% 662 | 94% 582 | 92% 201 | 85% 859 | 85% 546 |
| G4-S | - | 66% 1136 | 67% 324 | 60% 1280 | 60% 781 | - | 90% 563 | 94% 175 | 85% 739 | 85% 473 |
| C | - | - | 58% 343 | 54% 1341 | 55% 820 | - | - | 89% 141 | 85% 648 | 86% 414 |
| Author | - | - | 69% 49 | 65% 428 | 55% 93 | - | - | 87% 31 | 83% 262 | 76% 46 |
| Human | - | - | - | 63% 721 | 81% 892 | - | - | - | 81% 479 | 90% 631 |

(a) First Turn

| Setup | S1 (R = 33%) | | | | S2 (R = 50%) | | | |
|---|---|---|---|---|---|---|---|---|
| Judge | G4-S | Author | Human | Human-M | G4-S | Author | Human | Human-M |
| G4-P | 70% 1161 | 66% 341 | 66% 1325 | 68% 812 | 95% 727 | 88% 205 | 85% 864 | 85% 557 |
| G4-S | - | 65% 331 | 59% 1285 | 61% 783 | - | 89% 193 | 84% 776 | 85% 506 |
| Author | - | 67% 49 | 68% 413 | 63% 87 | - | 87% 31 | 86% 273 | 84% 54 |
| Human | - | - | 67% 707 | 83% 877 | - | - | 82% 474 | 91% 629 |

(b) Second Turn

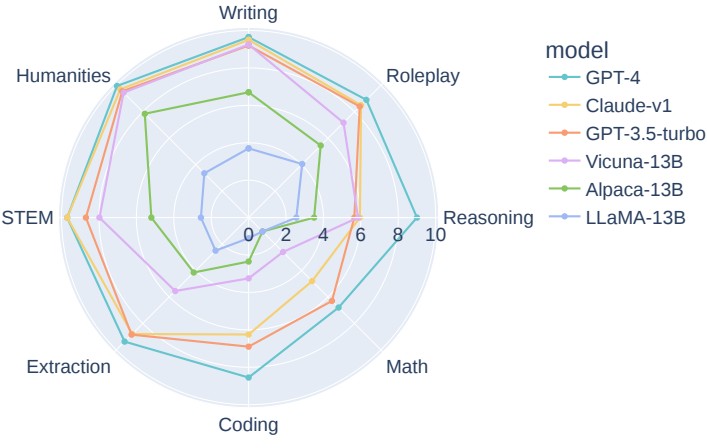

Figure 20: Category-wise scores of 6 models on MT-bench.

# E    Training Details of Vicuna Models

Vicuna is created by fine-tuning a LLaMA base model using user-shared conversations gathered from ShareGPT.com with its public APIs. ShareGPT is a website where users can share their ChatGPT conversations. To ensure data quality, we convert the HTML back to markdown and filter out some inappropriate or low-quality samples, which results in 125K conversations after data cleaning.[4] We then divide lengthy conversations into smaller segments that fit the model's maximum context length.

We construct three training datasets with different scales from this cleaned ShareGPT dataset. Their statistics are in Table 8, where we also compare it with Alpaca [38] dataset. "All" is the full dataset. "Single" only includes the first turn of each conversation. "Selected" is a small high-quality dataset of 3K sequences. To construct the "Selected" dataset, we pick sequences that include at least 3 turns of conversations generated by GPT-4 and run a clustering algorithm to divide them into 3K clusters and pick the centroid of each cluster.

All models (Vicuna-7B/13B) are trained with the same hyperparameters: global batch size=128, learning=2e-5, epochs=3, seq length=2048. Except for "Selected", which we train for 5 epochs. The training code is built on top of the Alpaca code but additionally handles multi-turn conversations. The training is done with 8x A100 GPUs. The longest single training run takes around 2 days. We utilize SkyPilot [49] managed spot instances for saving training costs and FlashAttention [11] for memory optimizations. The training code is available at `https://github.com/lm-sys/FastChat`.

Table 14: Dataset statistics

| Dataset Name | Alpaca | Selected | Single | All |
|---|---|---|---|---|
| #Token | 4.4M | 4.8M | 184M | 370M |
| #Sequence | 52K | 3K | 257K | 257K |
| Avg. turns of conversation | 1.0 | 4.0 | 1.0 | 2.9 |
| Avg. response length (token) | 65 | 343 | 473 | 373 |

---

[4]In this study, we use more data (125K) than the version in our earlier blog post (70K).

# F Exploring Vicuna as a judge

In this paper, we mostly evaluate the ability of close-sourced models such as GPT-4 as a proxy for human evaluations. However, model services such as GPT-4 can also become expensive with a growing number of evaluations. On the other hand, popular open-sourced LLMs, e.g. Vicuna-13B shows strong language understanding capability, and are much cheaper than close-sourced LLMs. In this section, we further explore the potential of using Vicuna-13B as a more cost-friendly proxy.

## F.1 Zero-Shot Vicuna

When using as-it-is (zero-shot), Vicuna-13B noticeably suffers from limitations we discuss, e.g. position bias. As shown in Table 15, Vicuna-13B has a consistency rate from 11.2% to 16.2% across different prompt templates, much lower than all the closed-sourced models. In addition, it has a high error rate (from 22.5% to 78.8%) because of its weaker instruction-following capability. In many scenarios, Vicuna-13B provides responses such as "Answer A is better than answer B", without following the pre-defined template. These responses are rendered as natural languages and are difficult to be parsed automatically, making the model less useful in a scalable and automatic evaluation pipeline.

## F.2 Arena Fine-tuned Vicuna

**Training**    Due to the incapability of the zero-shot Vicuna-13B model, we further finetune the model with human votes from Chatbot Arena. Specifically, we randomly sample 22K single-turn votes from the arena, covering all models supported by the time of this paper submission (GPT-4, GPT-3.5, Claude-v1, Vicuna-13b, Vicuna-7b, Koala-13B, Alpaca-13B,LLaMA-13B, Dolly-12B, FastChat-T5, RWKV-4-Raven, MPT-Chat, OpenAssistant, ChatGLM, and StableLM), to expose the model with a wider range of chatbot outputs and human preferences. We use 20K votes for training, and 2K for validation. To address the aforementioned weak instruction following problem, we formulate the problem as a 3-way sequence classification problem. Thus, the model simply needs to predict which one of the chat-bot outputs is better (or tie), without needing to exactly following the provided answer template. In particular, we construct an input by using the default prompt and the two model answers. The labels are A, B, and tie (including both-bad-vote and tie-vote). We train for 3 epochs with a cosine learning rate scheduler and a 2e-5 maximum learning rate. We use the 2K validation dataset to choose hyper-parameters, and test on the same 3K dataset in the main body of the paper.

**Position bias results**    The results for position bias are provided in Table 15. The consistency improves significantly from 16.2% to 65.0%. Due to the classification formulation, every output is recognizable (error rate 0%). In addition, we measure the classification accuracy over the test dataset.

**Agreement results**    It achieves 56.8% when including all three labels, and 85.5% when excluding tie predictions and labels, significantly outperforming random guesses of 33% and 50% respectively, and show positive signals to match GPT-4 (66% and 87% respectively). In conclusion, a further fine-tuned Vicuna-13B model shows strong potential to be used as a cheap open-sourced replacement for expensive closed-sourced LLMs. A similar conclusion is also found by a concurrent paper[42].

Table 15: Position bias of the Vicuna-13B model without and with further fine-tuning. We denote them as Vicuna-13B-Zero-Shot and Vicuna-13B-Fine-Tune respectively. Consistency is the percentage of cases where a judge gives consistent results when swapping the order of two assistants. "Biased toward first" is the percentage of cases when a judge favors the first answer. "Error" indicates wrong output formats. The largest number in each column is in bold.

| Judge | Prompt | Consistency | Biased toward first | Biased toward second | Error |
|-------|--------|-------------|---------------------|----------------------|-------|
| Vicuna-13B-Zero-Shot | default | 15.0% | **53.8%** | 8.8% | 22.5% |
| | rename | 16.2% | 12.5% | **40.0%** | 31.2% |
| | score | 11.2% | 10.0% | 0.0% | **78.8%** |
| Vicuna-13B-Fine-Tune | default | **65.0%** | 27.5% | 7.5% | 0.0% |

