# OpenReview forum: "Judging LLM-as-a-Judge with MT-Bench and Chatbot Arena"
_NeurIPS.cc/2023/Track/Datasets_and_Benchmarks — NeurIPS 2023 Datasets and Benchmarks Poster_

### Official Review · Reviewer_1xKM · 2023-07-18
**Paper review**

**Rating:** 7
**Confidence:** 4

**Strengths:**

* This paper shows that, in the near future, LLMs may be used to evaluate language models on advanced multi-turn language generation tasks. This comes at a much reduced cost compared to employing human evaluators.
* The authors carefully consider LLM biases in their evaluation method and introduce techniques to mitigate these biases. This list might not be exhaustive, but mitigating for these biases shows that LLMs show a high agreement rate with humans, which suggests that they are well calibrated judges.
* The paper introduces Chatbot Arena, a popular platform where public users can interact with LLMs. This platform generates good quality data providing plenty of human evaluations.

**Additional Feedback:**

No additional feedback.

**Clarity:**

The paper is well written, although there are quite a few typos and some sections feel like they were written in a hurry.

**Correctness:**

* There's quite a few results which are claimed to be included in the appendix, but the appendix is not available for review.
* The abstract says that the LLM as judge method is an "explainable" way of approximating human preferences. I don't see how this is explainable, especially since the paper exhibits some biases which are quite unpredictable (e.g. name preference in the "rename" experiment)

**Documentation:**

This is well documented.

**Ethics:**

No ethical concerns.

**Limitations:**

Limitations of the work are well discussed.

Another potential limitation to be discussed is what would happen if LLMs were to be fine-tuned on LLM judge preference data. Since the paper proposes to use LLMs instead of human evaluators, it is important to think about how the biases discussed in the paper could propagate during fine-tuning.

**Opportunities For Improvement:**

* It would be good to see more statistical analysis of the agreement between LLM as judge and human evaluators. The 80% agreement rate is often mentioned, but could we see more fine-grained data? Do LLMs disagree with humans on questions where humans typically disagree? Or is LLM/human disagreement independent from human/human disagreement?
* It would be good to try to find more subtle preference biases which may not be detectable at a purely syntactical level. Would it be possible to compare which LLM judges prefer which data points?
* Could we also see if the LLM/human disagreement rate is uniform across evaluation domains?

**Relation To Prior Work:**

This paper addresses prior work well. The paper acknowledges multi-turn conversational evaluation datasets, but argues that these datasets are too elementary. By introducing Chatbot Arena, this paper makes much more high quality evaluation data available.

**Summary And Contributions:**

This paper studies the feasibility of using LLMs to replace humans when evaluating language models on fine-grained NLP tasks. The paper introduces two new benchmarks which make it possible to evaluate how much an LLM judge agrees with a typical human evaluator. The paper carefully considers LLM judge biases and possible ways of mitigating them. The authors evaluate several LLMs and find that GPT-4 can act as a typical human evaluator in many instances.

---

> ### Author Response · Authors · 2023-08-15
>
> We appreciate your feedback. Below are our responses to your questions.
>
> > It would be good to see more statistical analysis of the agreement between LLM as judge and human evaluators. Do LLMs disagree with humans on questions where humans typically disagree?.
>
> Yes. In Figure 2, we observe the agreement between GPT-4 and humans progressively increases in line with the performance disparity of the model pairs (i.e., larger win rate difference). We will add more results.
>
> > Could we also see if the LLM/human disagreement rate is uniform across evaluation domains?
>
> The non-tie agreement rates on different MT-bench domains are mostly uniform (85%-88%) but with two outliers. The math domain has the highest agreement (96%) and the wiring domain has the lowest agreement (72%). This is reasonable because writing is relatively subjective and math is relatively objective.
>
> > but the appendix is not available for review.
>
> During the review, the appendix can be found in “Supplementary Material”. To make it easier for you to find it, we updated our pdf to directly include the appendix. Now you can download our updated PDF and see the appendix.
>
> > I don't see how this is explainable,
>
> In Figure 1. The LLM judge does not only output which is better but also includes an explanation of its judgment.

---

> > ### Comment · Reviewer_1xKM · 2023-08-28
> >
> > Thank you for your response.
> > >In Figure 1. The LLM judge does not only output which is better but also includes an explanation of its judgment.
> >
> > Thank you for the clarification regarding explainability. I would argue that the model's explanation is not necessarily a good source of explanation for the judgment. Your work exhibits biases regarding the ratings which clearly show that the model's explanation of its own judgments are partial at best (e.g. the model won't say  "I preferred A over B because I usually prefer the first of any two options"). It might be good to bring this out in the final version of the paper.

---

### Official Review · Reviewer_QiZU · 2023-07-21
**Open-ended set of tasks to evaluate LLMs abilities, and evaluating LLM as performances scorer for these**

**Rating:** 7
**Confidence:** 4

**Strengths:**

- Dataset of open-ended tasks is needed, the tasks included seem valid. Current LLM evaluation benchmark usually contains only a limited number of such tasks. The dataset has the potential to be used for other case studies not included in the paper.
- using an off-the-shelf LLM as a judge is now very popular in various of tasks, yet how to evaluate performances is not fully clear. The paper opens the doors on how to evaluate such.
- Comprehensive discussion about the biases that LLMs as a judge possess, and simple solutions to mitigate this, as well as evaluation of different experimental design choices.
- Comprehensive evaluation of of-the-shelf LLMs judgments and the agreement with humans
- Comprehensive evaluation of of-the-shelf LLMs performances in solving the suggested tasks
- Discussion of how to fine-tune LLM (vicuna) to serve as a judge (although only brief and is not included in other comparisons, see comments below).


**Additional Feedback:**

Please refer to all the sections above


**Clarity:**

Overall the paper is clear.
Other than the clarity issues already mentioned, I advise the author to refine the explanation of table 5: “The agreement between two random judges under each setup is denoted as “R=”‘: which judges? This doesn’t align with the numbers for G4 and human in the table, are more judges included here?


**Correctness:**

Paper statement and evaluation seems fine to me. My only “correctness” concerns is with cleaning the crowdsourcing data, as mentioned above.

**Documentation:**

Dataset is seems to be separated across different platforms (google drive, github, hugging faces). Will be easier for the user to refer to a single source.

Authors didn't include the required datasheet for dataset

**Ethics:**

no concerns

**Limitations:**

- Other than releasing the dataset of tasks, the authors should include the benchmark, that is, the row evaluations of the judge LLMs. This is crucial.
- The MT set contains only 80 controlled tasks. This is quite a few. What are the challenges in increasing this to a few hundredth? This will make the dataset much stronger. For example the biases of judge LLMs are calculated on relatively small dataset.
- Cleaning crowdsourcing data: crowdsourcing responses are known to be noisy, yet it seems like a cleaning process has been done on this part of the dataset. Authors should consider how to include such a process to eliminate any bias in their data. Especially since in contrast to the MT set, the Chatbot arena only includes a single human judment per question.


**Opportunities For Improvement:**

- Title is misleading, not clear how “vicuna evaluation” is related.

- The paper mentions fine-tuning vicuna to serve as the judge, but then only includes it in the appendix, with only a partial evaluation. Will be nice to repeat all the human agreement experiments as done for gpt-4 (tables 5,6,7). This will make the paper much stronger, mainly convincing that the fine-tuned vicuna judgments are equivalent to gpt-4 judgment and follow human judgments. Furthermore the fine-tune model is not included in the dataset. Releasing the judge model itself will make the contribution stronger. Especially concerning the reproducibility of GPT-4 judgments.

- The paper contains several scoring methods for comparing performances. In general it will be good to increase the clarity of those by giving actual examples for calculators or better textual explanations. For example, it is currently unclear to me how the scores of table 2 are calculated (I would expect bias left + bias right = 100%, but this is clearly not the case, how?).

- This is a dataset paper, yet there are only a few examples for the tasks included (table 1). And these are only from the MT set. Clearer visualizations and more examples of the dataset are needed.

- Paper evaluates the agreement between different judges, it will be very interesting to measure the agreement between a model to itself.


**Relation To Prior Work:**

Authors mentions existing datasets and benchmarks, and clearly states how the new dataset differs from these

**Summary And Contributions:**

The paper suggests a new dataset of open-ended tasks to evaluate LLMs capabilities on multiple domains. The dataset of tasks is collected both in a control way and through a crowd sourcing platform. The authors then assess the abilities of different judge LLMs to evaluate the performances of their peer LLMs on solving these tasks in different settings (e.g. pairwise comparison, single answer), and compare their evaluation with those of human raters. GPT-4-judge seems to highly agree with human ratings.

---

> ### Author Response · Authors · 2023-08-15
>
> We appreciate your feedback. Below are our responses to your questions.
>
> > Title is misleading, not clear how “vicuna evaluation” is related.
>
> We have removed “vicuna” from our title. The new title is “Judging LLM-as-a-judge with MT-Bench and Chatbot Arena”. Please see our updated paper.
>
> >Releasing the judge model itself
>
> The vicuna judge is only a three-way classifier. It is not as powerful as GPT-4 which can also generate an explanation. We plan to work on a judge model that can generate explanations as well before releasing them.
>
> > .. giving actual examples for calculators or better textual explanations. For example, it is currently unclear to me how the scores of table 2 are calculated (I would expect bias left + bias right = 100%, but this is clearly not the case, how?).
>
> In Table.2, Consistency + bias left + bias right + error = 100%. We will add more explanations and release more code to improve clarity.
>
> > Clearer visualizations and more examples of the dataset are needed.
>
> Thanks for the suggestions. We host several online interactive demos and datasets at the following URLs:
> MT Bench question and answers:  https://huggingface.co/spaces/lmsys/mt-bench/tree/main
> Leaderboard:  https://huggingface.co/spaces/lmsys/chatbot-arena-leaderboard
> Dataset sample viewer: https://huggingface.co/datasets/lmsys/mt_bench_human_judgments
> Dataset sample viewer 2: https://huggingface.co/datasets/lmsys/chatbot_arena_conversations
>
> We will include more of them in the paper.
>
> > ..it will be very interesting to measure the agreement between a model to itself.
>
> We already did some similar experiments. In Table.5 and Table. 6, we compute the agreement between GPT-4 single and GPT-4 pair, which are from the same model. In Table. 2, we compute the consistency of a single judge after swapping the position of two answers. This can also be viewed as the agreement between a model to itself.
>
> > the authors should include the benchmark, that is, the row evaluations of the judge LLMs
>
> We released a lot of GPT-4 judgments in https://huggingface.co/datasets/lmsys/mt_bench_human_judgments/viewer/lmsys--mt_bench_human_judgments/gpt4_pair
> and https://huggingface.co/spaces/lmsys/mt-bench/tree/main/data/mt_bench/model_judgment
>
> They are all generated by our code at https://github.com/lm-sys/FastChat/tree/main/fastchat/llm_judge
>
> > The MT set contains only 80 controlled tasks.
>
> Good suggestion. Initially, we use a small set of 80 questions to make it easy for quality control and human review. We are working on some ideas to automatically generate more questions.
>
> > “The agreement between two random judges under each setup is denoted as “R=”‘: which judges?
>
> The two judges are “random judge”. A “Random judge” is a judge that randomly guesses the outputs. For example, if we count tie, then there will be three outcomes (i.e. A is better, B is better, tie). Then two random guessing judge has an agreement of 33%. Should we rename “random judge” to “random guessing judge” for better clarity?
>
> > Will be easier for the user to refer to a single source.
>
> We will use this URL https://github.com/lm-sys/FastChat/tree/main/fastchat/llm_judge as the landing page and put all links under it.

---

> > ### Comment · Reviewer_QiZU · 2023-08-22
> > **Thank you for the response**
> >
> > I appreciate the response.
> > It seems to me that one of the meaningful contributions of the paper is the fine-tuned vicuna judge model. It will be useful for many future works, securing reproducibility that open-ai models like GPT-4 are missing. The fact that the model is free and open source also makes it very appealing. As such it would be great to release this model, and I'm a bit disappointed with author replay about this issue. I'd like to encourage the authors to reconsider this, or correct me if I missing something. Not releasing the model will hurt the quality of the paper and I'll need to reconsider my ratings.

---

> > > ### Author Response · Authors · 2023-08-22
> > >
> > > Hi reviewer,
> > >
> > > We have released the weights of the fine-tuned judge model here: https://huggingface.co/lmzheng/fine-tuned-judge.
> > > Initially, we planned to release a stronger version of the model after doing more developments. However, given the interest and feedback, we now directly release the checkpoint we used in the paper.

---

### Official Review · Reviewer_Zr4f · 2023-07-21
**A interesting  work about LLM evaluation**

**Rating:** 6
**Confidence:** 4
**Correctness:** Yes
**Clarity:** Yes

**Strengths:**

1. The researchers have clearly laid out this well-structured paper, which includes a systematic and data-centric analysis of LLM-as-a-judge, making it comprehensible to readers.
2. They have executed an exhaustive evaluation of GPT-4 and other high-functioning language models, enriching the pool of research on large language models.
3. The authors have devised a diverse range of assessment strategies covering both pairwise and single-wise scenarios.
4. It laudably elucidates LLM-as-a-judge's limitations and offers practical solutions to overcome these constraints.


**Additional Feedback:**

1. While GPT-4 might prove to be a worthy evaluator of Chatbots and potentially mirror human-like judgment, it may fall short in measuring the LLMs' reasoning capacities. Could you clarify this aspect?
2. You claim that the agreement without tie between GPT-4 and humans hits 85%, which is marginally higher than the agreement between humans at 81%, as indicated in line 242. Could you elaborate on how this agreement was calculated amongst different individuals?
3. As part of the single-wise evaluation, you instruct LLM to assign a score ranging from 1 to 10. How do we ensure the scoring function is consistently defined across each evaluation by the LLM?

**Documentation:**

Yes

**Ethics:**

No ethical concerns identified.


**Limitations:**

The authors have accurately identified the potential challenges and societal impact of employing LLMs as evaluators.

**Opportunities For Improvement:**

1. The evaluation question set in their benchmark only includes 160 questions. For an evaluation set intended for large language models, this scope might be deemed unsatisfactory.
2. Given their limited reasoning abilities, GPT-4 and other robust LLM are unable to satisfactorily address math and coding-related questions – an area where LLM-as-a-judge strategy falters.
3. The paper primarily concentrates on the helpfulness of Chat Assistants, but it needs to give equal weightage to other aspects such as harmlessness and honesty.


**Relation To Prior Work:**

Yes

**Summary And Contributions:**

This paper delves into the potential use of potent Large Language Models (LLMs), such as GPT-4, Claude-v1, and GPT-3.5, as a metric for judging other language models. The authors compare the LLM's judgments to human preferences and conclusively ascertain that the LLMs can match human tastes reasonably well, reaching over 80% agreement. Moreover, the authors generously agree to release their question set and voting data, as well as human preference data to the broader AI community.

---

> ### Author Response · Authors · 2023-08-15
>
> We appreciate your feedback. Below are our responses to your questions.
>
> > The evaluation question set in their benchmark only includes 160 questions.
>
> Initially, we use a small set of 80 questions to make it easy for quality control and human review. We are working on some ideas (MT-bench V2) to automatically generate more questions.
>
> > Given their limited reasoning abilities, GPT-4 and other robust LLM are unable to satisfactorily address math and coding-related questions. … It may fall short in measuring the LLMs' reasoning capacities. Could you clarify this aspect?
>
> According to our preliminary results, the reference-guided judge can greatly alleviate the limited reasoning ability if correct references are provided. However, we do acknowledge the current LLM cannot generate good reference solutions by themselves. One temporary solution is to let human experts provide reference solutions.
>
> > Could you elaborate on how this agreement was calculated amongst different individuals?
>
> Yes, these two agreement numbers (85% and 81%) seem unintuitive and confusing at first glance. Therefore, we have provided a detailed definition of agreement with illustrative examples in Appendix D.3 (see the updated paper pdf). We also introduced a new type of judge called “human-majority”. The upper bound of the agreement between GPT-4 and humans is the agreement between human-majority and humans.
>
> >How do we ensure the scoring function is consistently defined across each evaluation by the LLM?
>
> We did not apply any methods here. Instead, we empirically found the judgments generated by our prompts are very consistent, which is verified by our agreement calculations. We hypothesize that GPT-4 has a relatively stable internal rubric after its training and RLHF.

---

> > ### Comment · Reviewer_Zr4f · 2023-08-28
> > **Re: Official Comment by Authors**
> >
> > Thanks to the author's detailed reply.

---

### Official Review · Reviewer_BsjF · 2023-07-23
**Great contributions to LLM evaluation**

**Rating:** 8
**Confidence:** 5
**Correctness:** Claims are mostly correct.

**Strengths:**

1. Systematic analysis: authors present some very impressive systematic study and insights on LLM-as-a-Judge's algorithmic choices and potential limitations. While some of them have been discussed in previous work, it is for the first time they are clearly compared and experimented in a unified setting using GPT-4, with sufficient supporting results and materials.
2. Original idea on multi-turn evaluation: despite automatic assessment using LLMs have been heated practiced in the past few months, most of them are limited to single-turn evaluation. MT-Bench is definitely an original attempt to evaluate the challenging multi-turn dialogue capabilities of LLMs (though there are some limitations).
3. Extensive experiments: The third part I do appreciate is that Chat Arena and MT-Bench are evaluations that have been tried in wild and by relatively large number of real users. To lead such effort, which is crucial for AI systems to step out of laboratories, requires conquering much challenges in engineering and operations.

**Additional Feedback:**

See above.

**Clarity:**

The paper is mostly well-written. Though, I find some confusing bold annotations in Table 5. Why is 85% and 81% being bold, and 86% unbold?

**Documentation:**

Suffficent.

**Ethics:**

N/A.

**Limitations:**

See above for most of the limitations.

**Opportunities For Improvement:**

Though I appreciate this work, I identify some flaws and limitations that could be properly addressed.

1. Including more analysis on Chat Arena's distribution: one thing I am really concerned is about the property of Chat Arena's prompt distributions. Despite 30k is a relatively large number for an academic attempt, it might not cover comprehensive enough topics. And as far as I notice, most people who played Chat Arena are researchers from AI academia and industries, which is likely to form some biases in data distribution. I am wondering if authors could provide more thorough details and analysis on the related issues.
2. Multi-turn evaluation LLM-as-a-Judge: I notice that the writing have some inconsistency on MT-Bench Score's design between method description, experiments, and authors' final decision.
	+ In Table 5, for the first turn evaluation of MT-Bench, the authors compare G4-Pair and G4-Single. For second turn, only G4-Pair is reported and G4-Single is missing. However, it seems that MT-Bench finally still adopts G4-Single as the scoring method. How do you decide to choose G4-Single as scorer without experimenting it on second turn MT-Bench?
	+ In Section 3.5, authors use a large paragraph to discuss the prompt design problem for making G4-Pair correct in MT-Bench. However, finally it is G4-Single rather than G4-Pair being adopted. And from my view, the problem can be naturally avoid if adopting G4-Single.
3. LLM-as-a-Judge calibration: I am convinced by your work that LLM-as-a-Judge can *qualitatively* comparing different LLMs' performance (from Table 7's reporting). However, I am wondering if the score are calibrated to human evaluation, or in other words, can also qualitatively reflects the gap between LLMs as human would feel. For example, on MT-Bench the gap between GPT-4 and GPT-3.5 is 0.2---while the gap between Vicuna-7B and Vicuna-13B is 0.3. However, from both traditional benchmarks and my personal experience, the gap between GPT-4 and GPT-3.5 should be much more significant than that between Vicuna-7B and Vicuna-13B. I hope authors could provide some results of human evaluation on Table 7 and analyze the calibration property of this score.
4. Insufficient testing samples in MT-Bench: finally I have a concern on the limited number of samples in MT-Bench---it's only 80. Despite I agree on this work's method and evaluation framework's contribution, the too small size of the dataset could be a problem for its practical use and credibility. I strongly recommend authors to enlarge this testing set to cover more comprehensive topics.

**Relation To Prior Work:**

The work has clearly discussed its difference from previous works.

**Summary And Contributions:**

The paper introduces an important challenge: how to evaluate LLMs reliably and automatically. Authors systematically explore solutions in several dimensions, including human/LLM-as-a-judge, single/multi-turn evaluation, pairwise/single scoring, etc. Sufficient human evaluations are also conducted to analyze the agreement between automatic scoring methods and human scoring. In all this work presents a comprehensive view of LLM-as-a-judge status quo and provide valuable resources and experience for following works.

---

> ### Author Response · Authors · 2023-08-15
>
> We appreciate your feedback. Below are our responses to the “Opportunities For Improvement” mentioned by you.
>
> 1. We did a clustering analysis of 33K prompts from the Chatbot Arena dataset (https://huggingface.co/datasets/lmsys/chatbot_arena_conversations ). We use mpnet-base-v2 to compute the embedding of each prompt and run a K-means clustering algorithm to create 20 clusters. We then sample a few centroid prompts from each cluster and manually summarize the topic. Here are the number of samples and topics for the top 8 clusters. We plan to add more analysis later. Since the whole dataset is public, readers are also welcome to do their own investigation.
>
> | Topic | The number of samples |
> | --- | --- |
> | Ask for tips (How to .., What should I do if..) | 2254 |
> | Roleplay | 1324 |
> | Computer science questions (e.g., algorithm, machine learning, concept) | 1231 |
> | Business writing | 1175 |
> | Math/Riddle| 1109 |
> | Knowledge questions (What is ..) | 1074 |
> | Programming | 1072 |
> | Story writing | 1065 |
>
> 2. In our updated paper, we have added these results to Table. 5, which shows G4-single also matches very well for the second turn. These results were in the appendix in our first draft.
> There are different pros and cons of G4-single and G4-pair. We examine both in the main body. The method used in section 3.5 applied to both G4-single and G4-pair, although we mainly use G4-pair as examples.
>
> 3. Thanks for the suggestions! We are hosting an up-to-date leaderboard at https://huggingface.co/spaces/lmsys/chatbot-arena-leaderboard, where we show Elo ratings from human judges and MT-bench scores from GPT-4 side-by-side for 20+ models. We hope this can serve as a calibration of MT-bench scores.
>
> 4. Good suggestion. Initially, we use a small set of 80 questions to make it easy for quality control and human review. We are working on some ideas to automatically generate more questions.

---

> > ### Comment · Reviewer_BsjF · 2023-08-31
> >
> > Thanks for your responses.

---

### Decision · Program_Chairs · 2023-09-22

**Decision:**

Accept (Poster)

**Comment:**

This paper investigated a new area of using LLMs to judge the performance of other language models.  It developed benchmarks that can be used to evaluate these LLMs when they are used as a judge.  All reviewers feel positive about this work.  Please check the section of  Opportunities For Improvement carefully for additional improvement ideas.